# Body Schema Self-Awareness and Related Dream Content Modifications in Amputees Due to Cancer

**DOI:** 10.3390/brainsci11121625

**Published:** 2021-12-09

**Authors:** Alessandra Giordano, Michele Boffano, Raimondo Piana, Roberto Mutani, Alessandro Cicolin

**Affiliations:** 1Centro di Medicina del Sonno, Dipartimento di Neuroscienze, Università di Torino, 10126 Torino, Italy; roberto.mutani@unito.it (R.M.); alessandro.cicolin@unito.it (A.C.); 2SC Chirurgia Oncologica Ortopedica, AOU Città della Salute e della Scienza di Torino, Presidio CTO, 10126 Torino, Italy; mboffano@cittadellasalute.to.it (M.B.); rpiana@cittadellasalute.to.it (R.P.)

**Keywords:** body schema, body image, self-perception, amputation, cancer, sarcoma, dream, dream content

## Abstract

Purpose: the evaluation of body image perception, pain coping strategies, and dream content, together with phantom limb and telescoping phenomena in patients with sarcoma who underwent surgery for limb amputation. Material and Methods: consecutive outpatients were evaluated at T0 (within 3 weeks after surgery) and T1 (4–6 months after surgery) as follows: demographic and clinical data collection; the Groningen Questionnaire Problems after Arm Amputation; the West Haven-Yale Multidimensional Pain Inventory; the Body Image Concern Inventory, a clinical trial to identify telescoping; and a weekly diary of dreams. Dream contents were coded according to the Hall and Van de Castle coding system. Results: Twenty patients completed the study (15 males and 5 females, mean age: 53.9 ± 24.6, education: 7.8 ± 3.4). All subjects experienced phantom limb and 35% of them experienced telescoping soon after surgery, and 25% still after 4–6 months. Both at T0 and T1, that half of the subjects reported dreams about still having their missing limbs. At T1 the patients’ perceptions of being able to deal with problems were lower, and pain and its interference in everyday life were higher yet associated with significant engagement in everyday activities and an overall good mood. The dream content analysis highlighted that males were less worried about health problems soon after amputation, and women showed more initial difficulties that seemed to be resolved after 4–6 months after surgery. Conclusions: The dream content analysis may improve clinicians’ ability to support their patients during their therapeutic course.

## 1. Introduction

Over the years, many authors have defined body schema, body image, and body awareness in different ways. Body image is referred to the mental representation of the body and its physical features [1,2]. It is a conscious image or representation of the self, a subjective individual’s picture of their own body. The concept of body image is referred to the body as it is perceived in the immediate consciousness. It includes the conceptual construction of the body, and the emotional attitude and conscious feelings about the body itself [3]. The construction and the modification of the body’s internal representation are linked with the surrounding environment components as either cognitive, perceptual, affective, or behavioral [4]. The body schema is a non-conscious performance of the body [3] that is actively triggered by certain internal or environmental events and cues [5]. Indeed, in this performance the body acquires a certain organization or style in its relation with its environment. For example, it appropriates certain habitual postures and movements. The body schema is an active, operative performance of the body rather than an image of the existing parts of the body. “It is the body as it actively integrates it position and responses in the environment” [3].

The self-concept schema depends on “cognitive generalizations about oneself derived from past experiences”. Such experiences lead and guide the processing of self-related information acquired from social knowledge [6]. Self-schemas ‘‘reflect one’s core and affect laden assumptions or beliefs about the importance and influence of one’s appearance in life, including the centrality of appearance to one’s sense of self’’ [1,2].

Differences have been found among categories in determining self-schema. Men’s and women’s awareness levels of body parts are diverse. For men, body awareness has more permanence compared to women’s, meaning that they are more aware of their body parts. Additionally, left-handed people seem to be also more certain in perceiving their body senses, compared to right-handed people, while the latter showed less body divergence, meaning they have fewer clearness levels of images of different body parts [7]. Body awareness may be considered the conceptualization of body image or the complexity of the principles of body schema, body posture, and body position concerning spatial motions; it is the basis of personality [8].

Information about body image is differently interpreted by each individual, also concerning a potentially worrisome physical appearance [5]. Some people cognitively interpret information in accordance with their psychologically important body-related concerns, and, conversely, cognitive dysfunctions can lead to medical problems with a bidirectional relationship. Information processing can be dysfunctional when influenced by medical pathologies such as eating disorders and gender dysphoria [9,10], oncologic disease [11,12,13,14], HIV/AIDS [15,16], dermatological disturbances [17], burns [18], amputations [19,20,21,22], permanent scars [23] or interesting phenomena such as phantom limb [24], and telescoping [8,25]. Phantom limb sensations (PLS) are a well-known phenomenon [26,27] in which an amputee continues to experience a normally limbed body with the sensation that an amputated or missing limb is still part of the body. Telescoping is characterized by the experience of a gradual proximal migration or shortening of the amputated limb, as perceived by the patient.

These phenomena can significantly alter the function of the body and can affect the body image and, ultimately, psychosocial well-being [28] and the quality of life. For these reasons, understanding perceptual distortions in pathological conditions is important to addressing body image concerns and helping suffering people in having a content and productive life [3,29,30].

In amputees, phantom limb sensations and telescoping are widely present with most individuals experiencing some type of phantom phenomena at some point, post-amputation [26,27,31,32,33,34,35,36].

The phenomenon of the phantom limb may be considered as an alteration of the sensory-motor pathway. Its origins are not fully understood. The contributions of the central nervous system (CNS) and/or the peripheral nervous system (PNS) are still under debate. Recent findings suggest that both peripheral and central mechanisms, including neuroplastic changes in cortical neural circuits, can contribute to PLS and in particular to phantom limb pain (PLP) [37]. Currently, the most commonly suggested CNS theory is the cortical remapping theory (CRT), in which the brain is believed to respond to limb loss by reorganizing somatosensory maps [38], and the PNS may work in conjunction with the CNS to cause and maintain the persistence of PLP.

The human brain has the power to undergo plastic changes after an alteration in cortical circuits, which are, for a large part, responsible for the conscious awareness of the body, so-called body image. Can unconscious or less conscious processes, that contribute defining body schema, undergo similar adapting changes [39], and how can non-conscious awareness be investigated?

Dream content analysis may be considered an indicator of the internalization of the above-mentioned modifications.

A single definition of dream and dreaming is not possible. Dreaming is, at least in part, a mental experience that can be described during waking consciousness [40].

Dreaming should not be exclusively defined as a non-conscious electrophysiological state. Neurobiological theories of dreams state that the meaning of dreams should be considered clearly “transparent” and not elaborated [41,42]. Hobson affirmed that there are common formal aspects concerning everybody’s dreams, and there are some content differences related to age, gender, personality characteristics, and cross-cultural differences [43].

Domhoff hypothesized a continuity between dreams and waking life in the sense that “the concerns people express in their dreams are the concerns they have in waking life” [44,45]. Additionally, Revonsuo argued that dream content is not random, but organized and selective, and indicated that dream content is consistently and powerfully modulated by certain types of waking experiences [46].

Following Domhoff’s hypothesis, the analysis of dream reports may be useful in evaluating the changes of mental body image and self-perception in dreams after patients have experienced different kinds of disease-implicating body disfigurements. Calvin S. Hall and Robert L. Van de Castle [47,48] developed a specific coding system to study dream content. The Hall and Van de Castle system treats a dream report as a story or play in which there are several empirical categories.

Considering that both innate factors and life experiences may affect the body representation in dreams, they assumed that “the frequency with which a dream element appears reveals the concerns and interests of the dreamer”, providing the opportunity “to link dream content with the waking thoughts and behavior of the dreamer” [49].

Many studies evaluated the dream content in healthy volunteers to find common features in people’s dream content [44,45,49,50].

Women have been reported to recall their dreams more often than men. Sexual themes arise more often in men’s dreams, as do aggressions, whereas women’s dreams have more depressive contents [51,52].

Studies on dream content in amputated or surgically treated patients, analyzing the alterations of the physical aspect and their impact on the quality of life, mostly reported that a majority of amputees continue to dream of themselves with an intact body [39,53,54,55,56]. Our previous research demonstrated that dream content in patients who have recently undergone surgery for breast cancer is different both from healthy subjects dream content and from that of the same patients 3 months after surgery, especially for what concerns body image. This evidence underlines the importance of considering the timing of the assessment after amputation [14].

The aim of this study is the evaluation of body image perception, pain coping strategies, dream content, and their possible modifications over time, together with phantom limb and telescoping phenomena in patients affected by sarcoma who have undergone surgery for limb amputation, within six months after surgery.

## 2. Materials and Methods

Sleep medicine experts evaluated all consecutive outpatients referred to the SC Chirurgia Oncologica Ortopedica- Città della Salute e della Scienza over one year. Inclusion criteria were the diagnosis of bone or soft tissue sarcoma, minimum age 14 years old. Exclusion criteria were impaired cognitive ability (MMSE < 24), psychiatric disease, ray amputations (or smaller), or previous amputations.

Subjects were evaluated at T0 (within 3 weeks after surgery) and T1 (4–6 months after surgery). The evaluation included the following: the Groningen Questionnaire Problems after Arm Amputation (GQPAA), to describe frequency and type of phantom sensation, phantom pain, and stump pain, medical treatment received and their effect, and prosthetic use [26]; a weekly diary of dreams; and the West Haven-Yale Multidimensional Pain Inventory (WHYMPI), an inventory used in the assessment of clinical pain to examine the impact of pain on patients’ lives, the responses of others to the patients communications of pain, and the extent to which patients participate in common daily activities [57]. The WHYMPI is a 52-item, 12-scale inventory that is divided into three parts. Part I includes five scales designed to measure important dimensions of the chronic pain experience including (1) perceived interference of pain in vocational, social/recreational, and family/marital functioning, (2) support or concern from spouse or significant other, (3) pain severity, (4) perceived life control, and (5) affective distress. Part II assesses patients’ perceptions of the degree to which spouses or significant others display Solicitous, Distracting, or Negative responses to their pain behaviors and complaints. Part III assesses patients report of the frequency with which they engage in four categories of common everyday activities: Household Chores, Outdoor Work, Activities Away from Home, and Social Activities. An additional General Activity scale score, obtained from the combination of all four activity scale scores, has been added.

The Body Image Concern Inventory (BICI), a brief self-report measure of dysmorphic concern [58] a clinical trial to examine the location of the phantom limb useful to identify telescoping and changes in body schema [8]. The first evaluation of all patients also included demographic (age, instruction level, employment, family) and clinical (treatments, oncologic diagnosis) data collection

Dream contents were coded according to the Hall and Van de Castle (1966) coding system. Contents are summarized in the following output categories: Male/Female, Familiarity Friends Family Dead & Imaginary Animal Aggression/Friendliness Befriender Aggressor Physical Aggression Indoor Setting Familiar Setting, Self-Negativity, Bodily Misfortunes, Negative Emotions, Dreamer-Involved Success, Torso/Anatomy, Aggression, Friendliness, Sexuality, Misfortune, Good Fortune, Success, Failure, and Striving.

Dreams were recorded using a seven-day dream diary and blindly scored (mixing T0 and T1 dream reports) by a sleep medicine board-certified psychologist; difficult or ambiguous issues were resolved during in-group discussions. The alphanumeric codes were uploaded into a DreamSAT spreadsheet, which provided frequencies for the total series of dream reports and percentage calculations [49,59]. Coded dream data also were statistically compared to normative data using Adam Schneider’s DreamSAT (available online: www.dreamresearch.net, accessed on: 14 January 2021).

The Local Ethical Committee approved the protocol, and all patients (or parents whether under the age) signed the informed consent form.

Statistical analyses were carried out with IBM SPSS 27.0 for Windows, performing a Wilcoxon Test, the Chi-square test (X^2^), and contingencies tables. Results are presented in terms of mean ± SD with significance levels at *p* ≤ *0*.05.

## 3. Results

Twenty-seven subjects were enrolled. Seven out of 27 patients did not complete the T1 evaluation—two due to death, two did not show up at the appointment, and three were unable to attend the interview due to their disability. Twenty patients completed the study (15 males and 5 females). The mean age was 53.9 ± 24.6, education (years) 7.8 ± 3.4, and 10 patients had a diagnosis of soft tissue sarcoma and 10 of bone sarcoma. Eleven underwent lower limb amputation and 9 upper limb amputation, and pre-surgery pain was present in 13 out of 20 subjects. The mean number of weeks after amputation was at T0 2.6 ± 1.0 and at T1 19.5 ± 4.1.

Additional clinical data and Wilcoxon test results performed on BICI and WHYMPI are shown in Table 1.

All our subjects experience PLS at some point in their clinical history, and 1/3 of them experienced telescoping soon after surgery, with 1/4 still after 4–6 months. Both at T0 and T1, that half of the subjects reported dreams about still having their missing limbs.

Results showed significant differences at some West Haven-Yale Multidimensional Pain Inventory subscales.

Four-six months after surgery, significantly higher scores were obtained on two scales measuring important dimensions of how pain affects patient lives (interference and pain severity) and on patients’ reports of the frequency with which they engage in common everyday activities (house chores, activity away, social activity, and general activity).

Additionally, scales about patients perception of being able to deal with and control their problems (life control) and affective distress, and about the support or concern from spouse or significant other (support), obtained lower scores, along with patients’ perceptions of the degree to which significant others react and respond to them when they know they are in pain, and display negative response to their behaviors and complaints (negative response).

Crossing pre-surgery pain, amputation types and stump pain with telescoping both at T0 and T1, significance (*p* = 0.016), has only been found between pre-surgery pain and early post-amputation telescoping.

DreamSAT

One hundred and twenty-three dreams have been collected and codified using the Hall and Van de Castle method. Statistical analyses were performed by the automated Dream Data Entry System and statistical analysis tool DreamSat (spreadsheet for Microsoft Excel).

Crossing the number of reported dreams at T0 and T1 in both males and females showed that a statistically significant (*p* < *0*.05) higher number of dreams was reported by males (T0M/F = 44/18, T1M/F = 47/14) both at T0 and T1.

Table 2 and Table 3 show DreamSAT dream contents results for Males and Females vs. norms, both at T0 and T1, and Males and Females T0 vs. T1.

We found that males at both T0 and T1 had fewer negative features (such as aggression and negative emotions) and sexuality and more good fortune- and family-related content in their dreams compared to normative data. In addition, at T1 they showed no torso-anatomy references.

Females at T0 showed more familiar settings in bodily misfortune and dreamer-involved success, and less friends, friendliness, aggression, and failure, compared to norms. Besides, there were no torso-anatomy references.

Females, some months after surgery, showed less aggression, self-negativity, negative emotions, torso-anatomy content, success, and dreamer’s success compared to norms.

Males showed, substantially, no significant differences between T0 and T1, whereas women at T0 showed more bodily misfortune, aggression, and familiar settings.

## 4. Discussion

The present study analyzes the dream content in amputees, using the Hall and Van de Castle coding system, to understand the possible internalization of body modifications due to amputations and some interesting phenomena such as phantom limb and telescoping that may have significant implications on the quality of life. The results of this study mostly confirm previous literature data on phantom limb phenomenology: all our subjects experience PLS at some point in their clinical history, and 1/3 of them experienced telescoping soon after surgery, with 1/4 still after 4–6 months.

The pain inventory showed that, at T1, the patients’ perception of being able to deal with problems is lower and pain and its interference on everyday life is higher, meaning that physical sensation and phantom limb pain are still important issues also after 4–6 months from surgery. Nevertheless, the patients’ reports of the frequency with which they engage in four categories of common everyday activities (house chores, activity away, social activities, and general activities) is higher at 4–6 months after surgery, as they report an overall good mood. That is probably also because significant others display less negative responses to their pain complaints.

Besides, the significant association between pre-surgery pain and telescoping at T0 may mean that pre-surgery pain is predictive of telescoping soon after amputation.

In literature, we found suggestions that pre-amputation pain might predispose amputees to PLP [60]. Recent work has suggested that features related to the amputation, such as pre-amputation pain, stump pain, sleep disturbances, and/or diabetic/traumatic causes of amputation, may be associated with PLP development [35]. These data may support an association also between pre-surgery pain and telescoping, an association that, however, needs to be further evaluated.

In normal conditions, various factors may influence body image: Body mass index (BMI), family, peers, society, media, culture, self-esteem, gender, age, marital status, education level, smoking status, alcohol consumption, physical activity, weight control behavior, and spirituality [61,62]. For example, as we previously stated, for men, body awareness has more permanence compared to for women, meaning that they are more aware of their body parts. In addition, left-handed people seem to be also more certain in perceiving their body senses, compared to right-handed people, while the latter showed less body divergence, meaning that they have fewer clearness levels of images of different body parts [8].

Pathological conditions can have a significant impact on the body’s internal representation, as well. Indeed, body representation and self-perception rely on cognitive schema and emotional experiences that are strongly linked with what happens in everyday life.

For this reason, self-awareness can change during life due to life experiences and pathological conditions that can occur.

Although there has been much progress in the surgical treatment of bone and soft tissue sarcomas, amputation is unfortunately still needed in 5–10% of cases [63] of bone and soft tissue tumors.

As we previously said, phantom limb sensations are very common in amputees [8,26,31,32,33,34,35], and post-amputation pain such as phantom limb pain (PLP) and residual limb pain (RLP) may have negative effects on a person’s well-being.

Phantom limb sensations may include pain, itching, movements, abnormal shape or position, warmth, and cold or electric sensations. Some kind of phantom sensations are present in virtually all patients with limb amputation and may be explained by three theories: the peripheral one theorizes that the impulses carried to the central nervous system, coming from the nerve endings in the stump, are perceived as pertaining to the amputated limb [8,64]; the central theory affirms that visual sensory, tactile and postural impressions contribute to developing a life-long body schema independently from peripheral sensory impulses and define phantom limb as a conscious process [8,65,66]; and the mixed theory proposes that both these factors are combined [67]. Following these theories, the perception of the phantom limb phenomenon may be considered as an alteration of the sensory-motor pathway.

If PLS involve the majority of amputees and are still an important issue several months after amputation, as we found in our study, a different and deeper way to analyze the impact of these phenomena on patients life may be useful.

If, during waking life, an amputee still feels the presence of the amputated limb, what happens while dreaming? Can the dream be considered a reliable and precocious indicator of what happens in the dreamer’s emotional life?

Over the years, some studies have analyzed the dream content using a standardized coding method [51,68,69,70,71]. The continuity hypothesis of dreams suggests that the presence of aversive experiences in the waking state should be reflected in dream content [55,72].

In literature, most studies reported that a majority of amputees continue to dream about themselves with an intact body [36,45,46,47,48] even if, in some cases, there is a difference between how the phantom limb is perceived in the waking state and in dreams [73].

Bekrater-Bodmann and colleagues in 2015, demonstrated that, although only a minority of dreams in their sample contained bodily impairments, there is a positive relationship between post-amputation pain during wake and the recall of an impaired body representation in dreams [55]. Our previous research demonstrated that dream contents in subjects who have recently undergone surgery for breast cancer are different from that of healthy subjects and from that of the same patients three months after the surgery [14].

The dream analysis, in the present study, showed both at T0 and T1 that half of the subjects reported dreams about still having their missing limbs. As we previously said, this evidence underlines the importance of considering the timing of the assessment after amputation. Males both at T0 and T1 reported a statistically significantly higher number of dreams. That is maybe because, as we previously said, men’s body awareness has more permanence compared to women’s [74].

We also found that males, at both T0 and T1, had fewer negative features (such as aggression and negative emotions) and more good fortune and an abundance of family members in their dreams compared to normative data indicating fewer concerns about the surgery, the clinical outcome, and the physical appearance. In addition, at T1 they showed no torso-anatomy references, an indicator of possible disturbance of bodily image (denial) or a real absence of body-image issues.

Females at T0 showed more familiar settings, bodily misfortune, and dreamer-involved success, and fewer friends, friendliness, aggression, and failure compared to norms. Besides, there were no torso-anatomy references.

Female, some months after surgery, showed fewer aggression, self-negativity, negative emotions, torso-anatomy contents, dreamer’s success, and more good fortune compared to norms. Women’s self-perception appears less frightened, and dominated by negative emotions, maybe suggesting a good self-restructuring which could be tied to the ordeal of overcoming cancer.

Yet the complete absence of body-related topics in women’s dreams both at T0 and T1 could reflect clear neglect about the body-shape change.

T0 and T1 in males showed, substantially, no significant differences, whereas women at T0 showed more bodily misfortune, aggression, and familiar settings. According to Domhoff’s perspective, the presence of an abundance of family members and settings and increased misfortunes could be related to a dreamer’s maladjustment problems. As we previously said, these concerns seemed to be better managed some months after surgery.

It seems possible that males were less worried about the health problem from the beginning, and women showed more initial difficulties that seemed to be resolved after 4–6 months after surgery.

As explained by Revonsuo’s (2000) perspective and Ernest Hartmann’s (1996) theory, the function of dreaming is to “contextualize a dominant emotion of the dreamer”, making “connections more broadly than waking in the nets of mind” [14,75]. All these considerations are in agreement with Domhoff’s assertion that “dreams express several key aspects of people’s conceptual systems” [44].

Some limits in our study should be underlined. Difficulties in recruitment in this research field resulted in a small sample of patients and consequent limited statistical analysis power. Additionally, the wider SD we obtained at T1 was because we tried to reevaluate the patients during medical follow-up visits that were differently distributed in a large period of time. It was impossible to reach the patients during an additional visit. This could have missed possible additional changes in the body perception at 4 and 6 months after the amputation.

We also did not consider the possible prosthesis reconstruction, which could affect some variables mostly regarding the subjective sensation of a possible return to “integrity”. Some of the subjects in our study refused prosthesis, some were not eligible, and some of them were not in the process yet, so we did not have enough data to collect and analyze.

In addition, considering the persistence, in half of the patients, of the amputated limb in their dreams, it would be important to further re-test the patients and plan an extended follow-up.

## 5. Conclusions

The analysis of dream content, which could reveal the expression of inner self-perception through a novel analysis method, could significantly improve psycho-oncologists’ and clinicians’ ability to support their patients during their therapeutic course. The persistence of an intact self-image, even 4–6 months after surgery, in some of our patients’ dreams may suggest an ongoing plasticity process that has to be better evaluated to implement a timely intervention. Indeed, providing proper psychological support is essential in patients who have to face important challenges in self-perception and possible significant changes in the quality of their lives.

## Figures and Tables

**Table 1 brainsci-11-01625-t001:** Clinical data percentage and Wilcoxon Test results between T0 and T1 questionnaires subscales.

	T0 %	T1 %			
PLS ^§^	100	100			
Stump pain	40	45			
Telescoping	35	25			
Limb in dreams	50	50			
	Mean T0	Std. Dev. T0	Mean T1	Std. Dev. T1	Sign.
BICI ^§§^_	30.40	13.26	37.65	15.78	0.052
WHY ^§§§^_interference	0.87	0.86	1.70	1.62	0.0009 *
WHY_support	3.85	1.71	3.43	1.90	0.006 *
WHY_pain_sev	1.96	1.37	2.46	1.64	0.002 *
WHY_life_contr	2.95	1.95	2.28	1.19	0.049 *
WHY_aff_distr	2.97	1.81	1.34	0.88	0.000 *
WHY_neg_resp	0.66	0.80	0.43	0.66	0.012 *
WHY_solic_resp	3.84	1.40	3.49	1.99	0.078
WHY_dist_resp	2.57	1.32	2.75	1.69	0.959
WHY_house_chores	1.40	1.57	2.21	2.12	0.039 *
WHY_out_work	0.36	0.90	0.50	0.75	0.064
WHY_act_away	1.11	1.37	1.91	1.39	0.017 *
WHY_soc_act	2.13	1.71	3.28	1.27	0.005 *
WHY_gen_act	1.18	1.13	2.11	1.15	0.005 *

* Statistically significant; ^§^ PLS: Phantom Limb Sensations; ^§§^ BICI: Body Image Concern Inventory; ^§§§^ WHY: West Haven-Yale Multidimensional Pain Inventory (WHYMPI) subscales.

**Table 2 brainsci-11-01625-t002:** Male at T0 vs. norms and Female at T0 vs. norms and Male at T1 vs. norms and Female at T1 vs. norms.

	Male T0	Male Norms	*p* vs. Male	Female T0	Female Norms	*p* vs. Female	Male T1	Male Norms	*p* vs. Males	Female T1	Female Norms	*p* vs. Female
^§^ **Characters**												
Male/Female	75%	67%	0.371	13%	48%	* 0.022	78%	67%	0.239	50%	48%	0.915
Familiarity	53%	45%	0.282	50%	58%	0.502	56%	45%	0.170	60%	58%	0.900
Friends	24%	31%	0.296	6%	37%	** 0.001	27%	31%	0.534	40%	37%	0.802
Family	22%	12%	* 0.049	44%	19%	* 0.033	24%	12%	* 0.036	13%	19%	0.533
Dead and Imaginary	2%	0%	0.254	6%	1%	0.230	5%	0%	* 0.041	6%	1%	0.221
Animal	6%	6%	0.941	6%	4%	0.793	5%	6%	0.668	12%	4%	0.240
^§^ **Social Interaction Percents**												
Aggression/Friendliness	57%	59%	0.829	71%	51%	0.274	55%	59%	0.693	0%	51%	** 0.001
Befriender	20%	50%	0.152	50%	47%	0.935	80%	50%	0.158	50%	47%	0.909
Aggressor	64%	40%	0.114	40%	33%	0.744	50%	40%	0.557	0%	33%	0.223
Physical Aggression	23%	50%	* 0.044	33%	34%	0.980	8%	50%	** 0.000	0%	34%	0.080
^§^ **Settings**												
Indoor Setting	38%	48%	0.345	50%	61%	0.580	37%	48%	0.187	67%	61%	0.737
Familiar Setting	73%	62%	0.340	100%	79%	* 0.034	81%	62%	0.085	63%	79%	0.315
^§^ **Self-Concept Percents**												
Self-Negativity	59%	65%	0.465	60%	66%	0.698	67%	65%	0.713	38%	66%	* 0.046
Bodily Misfortunes	48%	29%	0.097	100%	35%	** 0.000	19%	29%	0.295	33%	35%	0.953
Negative Emotions	36%	80%	** 0.002	50%	80%	0.192	77%	80%	0.758	43%	80%	* 0.036
Dreamer-Involved Success	50%	51%	0.959	100%	42%	* 0.016	40%	51%	0.412	0%	42%	* 0.016
Torso/Anatomy	20%	31%	0.436	0%	20%	* 0.041	0%	31%	** 0.000	0%	20%	** 0.006
^§^**Dreams with at Least One**:												
Aggression	20%	47%	** 0.000	22%	44%	* 0.047	17%	47%	** 0.000	00%	44%	** 0.000
Friendliness	20%	38%	* 0.012	11%	42%	** 0.002	19%	38%	** 0.005	21%	42%	0.096
Sexuality	0%	12%	** 0.000	0%	4%	0.112	0%	12%	** 0.000	0%	4%	0.159
Misfortune	30%	36%	0.367	22%	33%	0.296	38%	36%	0.776	21%	33%	0.319
Good Fortune	20%	6%	** 0.005	6%	6%	0.994	21%	6%	** 0.002	36%	6%	** 0.003
Success	7%	15%	0.089	11%	08%	0.614	13%	15%	0.672	0%	8%	* 0.039
Failure	11%	15%	0.450	0%	10%	** 0.008	19%	15%	0.515	21%	10%	0.229
Striving	18%	27%	0.178	11%	15%	0.658	28%	27%	0.923	21%	15%	0.515

^§^ In bold: DreamSAT output categories; * Statistically Significant; ** Highly statistically significant.

**Table 3 brainsci-11-01625-t003:** Male T0 vs. T1 and Female T0 vs. T1.

	Males T0	Males T1	*p*: Males T0 vs. Males T1	Females T0	Females T1	*p*: Females T0 vs. Females T1
^§^ **Characters**						
Male/Female	75%	78%	0.784	13%	50%	0.166
Familiarity	53%	55%	0.855	50%	64%	0.481
Friends	24%	28%	0.747	6%	55%	** 0.003
Family	22%	23%	0.995	44%	0%	** 0.000
Dead and Imaginary	2%	2%	0.892	6%	9%	0.721
Animal	6%	5%	0.807	6%	0%	0.214
^§^ **Social Interaction Percents**						
Aggression/Friendliness	57%	55%	0.894	71%	0%	0.060
Befriender	20%	80%	* 0.042	50%	100%	0.200
Aggressor	64%	50%	0.552	40%	0%	0.211
Physical Aggression	23%	8%	0.262	33%	0%	0.254
^§^ **Settings**						
Indoor Setting	38%	39%	0.933	50%	71%	0.426
Familiar Setting	73%	83%	0.528	100%	67%	* 0.042
^§^ **Self-Concept Percents**						
Self-Negativity	59%	66%	0.565	60%	50%	0.734
Bodily Misfortunes	48%	20%	0.078	100%	0%	** 0.000
Negative Emotions	36%	78%	0.054	50%	25%	0.459
Dreamer-Involved Success	50%	63%	0.640	100%	25%	0.087
Torso/Anatomy	20%	0%	0.060	0%	0%	1.000
^§^**Dreams with at Least One**:						
Aggression	20%	24%	0.693	22%	0%	* 0.013
Friendliness	20%	27%	0.486	11%	10%	0.927
Sexuality	0%	0%	1.000	0%	0%	1.000
Misfortune	30%	36%	0.528	22%	20%	0.890
Good Fortune	20%	18%	0.803	6%	10%	0.671
Success	7%	15%	0.239	11%	0%	0.085
Failure	11%	9%	0.744	0%	0%	1.000
Striving	18%	18%	1.000	11%	0%	0.085

^§^ In bold: DreamSAT output categories; * Statistically Significant; ** Highly statistically significant.

## Data Availability

The data presented in this study are available on request from the corresponding author.

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
