# Peer review of "Body Schema Self-Awareness and Related Dream Content Modifications in Amputees Due to Cancer"

_brainsci, 2021, doi:10.3390/brainsci11121625_

Round 1
Reviewer 1 Report
In this empirical research contribution, the authors analyzed a variety of constructs pertaining to the bodily self and related them to dream contents in individuals who underwent amputation due to cancer. The assessment was performed at two different time points (within 3 weeks and 4-6 months after surgery) Well-established questionnaires and tests have been administered for this purpose (i.e., Hall and Van de Castle coding system). A particular focus was on the impact on the quality of life.
The research ideas behind the paper are of general interest, and the methodological approach overall is correct. However, it is the feeling of the reviewer that the report needs fundamental (i.e., structural changes) to be suitable for publication.
Generally, the manuscript needs a more accurate definition of the proposed constructs and how they are measured (for example, the Hall and Van De Castle coding system should be defined in all its subcomponents.
Also, once a construct is defined, terminology to refer to it is not always consistent. Moreover, many of the descriptions proposed in the discussion may better fit in the introduction
The more specific concerns are addressed below, following the article’s flow.
Simple summary
- Body Image and body schema are introduced, but they seem to be used interchangeably in the simple summary. Please provide a brief definition of body schema and how the construct separates from body image
Abstract
- Please report demographic information on the tested individuals with an amputation (age, sex, sd)
- Line 28: What kind of difficulties do they experience.? Consider making this point clearer.
Introduction
- 63: Consider specifying which kind of alteration of the sensorimotor pathway occur
- 64-65: Body schema and body Image are used as synonymous; this sounds arbitrary and may need more clarifications. It could be helpful to provide an example of conscious or non-conscious awareness
- 89: The provided references are fairly backdated. Consider adding a citation (Saetta et al., 2020, Nature: Scientific Data) for a most recent account
Materials and Methods:
- 109: The subscales of the Haven-Yale Multidimensional Pain Inventory could be briefly explained. This would also make the table reading much easier
- 129: Consider reporting median and interquartile ranges since the analyses performed include a Wilcox Test. Alternatively, should be data normally distributed, the implementation of T-test statistics might be more opportune
Results
-134: The analyses of the differences between upper and lower limb amputees may provide fascinating insights into to understand the impact of different limbs loss on the bodily self and the quality of life
-137: The standard deviation number of weeks after amputation is different at T0 and T1. Why did the authors collapse the data over two months? Would they not expect the presence of additional changes in bodily self 4 and 6 months after the amputation? Please clarify
- 165: Results about gender differences come as a surprise. Stating the rationale of such analysis already in the introduction could improve the readability of the paper.
- Small typo in Table 2 (168): “ale T0” ïƒ “Male T0”
Discussion:
- 193: Please specify the four categories of the ordinary everyday activities
- 197: The authors may consider elaborating on the relationship between presurgery pain and telescoping
- 198-200:After introducing “body schema, body image, and body awareness”, the authors focus on self-concept schema. These constructs could be introduced in the introduction (in line also with the comments above)
213: Please define body permanence and body divergence. Also, only after reading the discussion, the reader gets aware of the rationale of gender-effect analysis.
Author Response
REVIEWER 1
Corrections are in RED.
In this empirical research contribution, the authors analyzed a variety of constructs pertaining to the bodily self and related them to dream contents in individuals who underwent amputation due to cancer. The assessment was performed at two different time points (within 3 weeks and 4-6 months after surgery) Well-established questionnaires and tests have been administered for this purpose (i.e., Hall and Van de Castle coding system). A particular focus was on the impact on the quality of life.
The research ideas behind the paper are of general interest, and the methodological approach overall is correct. However, it is the feeling of the reviewer that the report needs fundamental (i.e., structural changes) to be suitable for publication.
Generally, the manuscript needs a more accurate definition of the proposed constructs and how they are measured (for example, the Hall and Van De Castle coding system should be defined in all its subcomponents).
Also, once a construct is defined, terminology to refer to it is not always consistent. Moreover, many of the descriptions proposed in the discussion may better fit in the introduction.
The more specific concerns are addressed below, following the article’s flow.
Simple summary
- Body Image and body schema are introduced, but they seem to be used interchangeably in the simple summary. Please provide a brief definition of body schema and how the construct separates from body image.
Thank you for your suggestion: we added: “conscious” and “The body schema is a non-conscious performance of the body that acquires a certain organization or style in its relation with its environment” and substituted the word “schema” with the word “image”.
We also changed the word “impact” with “affect”.
Abstract
- Please report demographic information on the tested individuals with an amputation (age, sex, sd)
Thank you for your suggestion. We added: “15 males and 5 females, mean age: 53.9±24.6, education: 7.8±3.4” in the abstract.
- Line 28: What kind of difficulties do they experience.? Consider making this point clearer.
Thank you for your suggestion: From the dream content analysis, it is not possible to analyze specific content from each subject. Reported topics and issues (from the dream diary) are summarized in general topics (such as health problems) that refer to main categories.
Introduction
- 63: Consider specifying which kind of alteration of the sensorimotor pathway occur
Thank you for your suggestion. We added: “Its origins are not fully understood. The contributions of the central nervous system (CNS) and/or the peripheral nervous system (PNS) are still under debate. Recent findings suggest that both peripheral and central mechanisms, including neuroplastic changes in cortical neural circuits, can contribute to PLS and in particular to phantom limb pain (PLP). Currently, the most commonly suggested CNS theory is the cortical remapping theory (CRT), in which the brain is believed to respond to limb loss by reorganizing somatosensory maps and the PNS may work in conjunction with the CNS to cause and maintain the persistence of PLP.
- 64-65: Body schema and body Image are used as synonymous; this sounds arbitrary and may need more clarifications. It could be helpful to provide an example of conscious or non-conscious awareness
Thank you for your suggestion.
We changed the introduction in order to be more clear, as follows: Over the years, many authors have defined body schema, body image, and body awareness in different ways. Body image is referred to the mental representation of the body and its physical features [1,2]. It’s a conscious image or representation of the self, It is the a subjective individuals’ picture of their own body. irrespective of how their body actually looks The concept of the body is referred to the body as it is perceived in the immediate consciousness. It includes the conceptual construction of the body, and the emotional attitude and conscious feelings about the body itself [3]
Also, we deleted: “body scheme, or” and added the following: “that contribute defining body schema”
- 89: The provided references are fairly backdated. Consider adding a citation (Saetta et al., 2020, Nature: Scientific Data) for a most recent account
Thank you for your suggestion. We added the following citation: “Saetta G, Cognolato M, Atzori M, et al. Gaze, behavioral, and clinical data for phantom limbs after hand amputation from 15 amputees and 29 controls. Sci Data. 2020;7(1):60. Published 2020 Feb 20. doi:10.1038/s41597-020-0402-1” [36]
We also added The Hall/Van De Castle system treats a dream report as a story or play in which there are several empirical categories.
Materials and Methods:
- 109: The subscales of the Haven-Yale Multidimensional Pain Inventory could be briefly explained. This would also make the table reading much easier
Thank you for your suggestion. We added the following in material and methods: “The WHYMPI is a 52-item, 12-scale inventory that is divided into three parts. Part I includes five scales designed to measure important dimensions of the chronic pain experience including; 1) perceived interference of pain in vocational, social/recreational, and family/marital functioning, 2) support or concern from spouse or significant other, 3) pain severity, 4) perceived life control, and 5) affective distress. Part II assesses patients’ perceptions of the degree to which spouses or significant others display Solicitous, Distracting or Negative responses to their pain behaviors and complaints. Part III assesses patients’ report of the frequency with which they engage in four categories of common everyday activities; Household Chores, Outdoor Work, Activities Away from Home, and Social Activities. An additional General Activity scale score, obtained from the combination of all four-activity scale scores, has been added”.
We also added: “Contents are summarized in the following output categories: Male/Female, Familiarity Friends Family Dead & Imaginary Animal Aggression/Friendliness Befriender Aggressor Physical Aggression Indoor Setting Familiar Setting, Self-Negativity, Bodily Misfortunes, Negative Emotions, Dreamer-Involved Success, Torso/Anatomy, Aggression, Friendliness, Sexuality, Misfortune, Good Fortune, Success, Failure, and Striving”.
- 129: Consider reporting median and interquartile ranges since the analyses performed include a Wilcox Test. Alternatively, should be data normally distributed, the implementation of T-test statistics might be more opportune
Since the Kolmogorov–Smirnov test showed a not-normal distribution, we performed a Wilcoxon test on our data. FYI, we attached the median and interquartile data you required that we decided not to put in the paper in order to make the paper reading easier.
Results
-134: The analyses of the differences between upper and lower limb amputees may provide fascinating insights into to understand the impact of different limbs loss on the bodily self and the quality of life
Thank you for your suggestion: “We performed a Mann-Whitney test correlating BICI and Haven-Yale Multidimensional Pain Inventory with upper and lower amputations. A significant difference has only been found between amputation and BICI at T0. We found higher values of BICI at T0 in lower limb amputees meaning that dysmorphic concerns and related behaviors (i.e. finding a way to hide the amputation) are higher soon after surgery. In Literature we only found that persistent pre-operative pain, proximal site of amputation, stump pain, lower limb amputation and phantom sensations were identified as risk factors for PLP (i.e. Limakatso K, Bedwell GJ, Madden VJ, Parker R. The prevalence and risk factors for phantom limb pain in people with amputations: A systematic review and meta-analysis. PLoS One. 2020;15(10):e0240431. Published 2020 Oct 14. doi:10.1371/journal.pone.0240431).
Since all our patients showed PLS, both at T0 and T1, we think we probably need a bigger sample to dig deeper into this relationship.
-137: The standard deviation number of weeks after amputation is different at T0 and T1. Why did the authors collapse the data over two months? Would they not expect the presence of additional changes in bodily self 4 and 6 months after the amputation? Please clarify
Thank you for your suggestion: the wider SD at T1 because we tried to reevaluate the patients during a medical follow-up visit that were differently distributed in a large period of time. It was impossible to reach the patients during an additional visit.
- 165: Results about gender differences come as a surprise. Stating the rationale of such analysis already in the introduction could improve the readability of the paper.
Thank you for your suggestion. We added the following in the introduction: “Women have been reported to recall their dreams more often than men. Sexual themes arise more often in men’s dreams, as does aggression whereas women’s dreams have more depressive contents (Schredl M, Sahin V., Schäfer, G. Gender differences in dreams: do they reflect gender differences in waking life?, Personality and Individual Differences, 1998, 25, 3, 433-442, https://doi.org/10.1016/S0191-8869(98)00035-X. Mathes, J., & Schredl, M. Gender differences in dream content: Are they related to personality? International Journal of Dream Research, 2013, 6, 2)
- Small typo in Table 2 (168): “ale T0” à “Male T0”
We modified the Table.
Discussion:
- 193: Please specify the four categories of the ordinary everyday activities
Thank you for your suggestion. We added: (house chores, activity away, social activities and general activities)
- 197: The authors may consider elaborating on the relationship between pre-surgery pain and telescoping
Thank you for your suggestion. We added: “In Literature, we found suggestions that pre-amputation pain might predispose amputees to PLP [60]. Recent work has suggested that features related to the amputation, such as pre-amputation pain, stump pain, sleep disturbances and/or diabetic/traumatic causes of amputation, may be associated with PLP development. These data may support an association also between pre-surgery pain and telescoping, an association that, however, needs further evaluation.
- 198-200: After introducing “body schema, body image, and body awareness”, the authors focus on self-concept schema. These constructs could be introduced in the introduction (in line also with the comments above)
Thanks for the suggestion. We anticipated the following in the introduction: “Indeed, in this performance the body acquires a certain organization or style in its relation with its environment. For example, it appropriates certain habitual postures and movements. The body schema is an active, operative performance of the body rather than an image of the existing parts of the body. “It is the body as it actively integrates it position and responses in the environment”
The self-concept schema depends on “cognitive generalizations about oneself derived from past experiences. Such experiences lead and guide the processing of self-related information acquired from social knowledge. Self-schemas ‘‘reflect one’s core, affect laden assumptions or beliefs about the importance and influence of one’s appearance in life, including the centrality of appearance to one’s sense of self’’
And: Appearance self-schemas are actively triggered by certain internal or environmental events and cues and Differences have been found among categories in determining self-schema. Men’s and women’s awareness levels of body parts are diverse. For men, body awareness has more permanence compared to women’s, and, in addition, left-handed people seem to be also more certain in perceiving their body senses, compared to right-handed, while the latter showed less body divergence [7]. Body awareness may be considered as the conceptualization of body image or the complexity of the principles of body schema, body posture, and body position concerning spatial motions; it’s the basis of personality
And (see also reviewer 2): “Dream should not be exclusively defined as a non-conscious electrophysiological state. Neurobiological theories of dreams stated that the meaning of dreams should be considered clearly "transparent" and not elaborated. Hobson affirmed that there are common formal aspects concerning everybody's dreams, and there are some content differences related to age, gender, personality characteristics, and cross-cultural differences”.
213: Please define body permanence and body divergence. Also, only after reading the discussion, the reader gets aware of the rationale of gender-effect analysis.
Thanks for your suggestion. We anticipated the following in the introduction: “and differences have been found among categories. Men’s and women’s awareness levels of body parts are diverse. For men, body awareness has more permanence compared to women’s, and, in addition, left-handed people seem to be also more certain in perceiving their body senses, compared to right-handed, while the latter showed less body divergence”.
And we specified in the discussion as follows: “For example, as we previously stated, for men, body awareness has more permanence compared to women’s, meaning that they are more aware of their body parts. In addition, left-handed people seem to be also more certain in perceiving their body senses, compared to right-handed, while the latter showed less body divergence meaning they have fewer clearness levels of images of different body parts”.

Reviewer 2 Report
The underlying topic is interesting, but I find the study design as well as the processing to be in need of improvement.
As the authors themselves describe in the study limitations, the sample is very small (if gender differences are taken into account, only 5 women were included in the study). Also, the study would have benefited from including the use or fitting of a prosthesis.
The QoL mentioned in the abstract could have been collected more extensively to draw better conclusions and also the statement that the dream content analysis can serve to improve the quality of treatment, I find no further mention in the current text.
In the discussion there are wide passages, which should be part of the Introduction.
Also methodologically it is not clear to me how the dream content and the phenomena phantom limb and telescoping are mixed up in this article.
More detailed analyses of the dream content, possibly with examples, as well as the conclusions drawn from them, would be of added scientific value.
Author Response
REVIEWER 2
CORRECTIONS ARE IN BLUE
The underlying topic is interesting, but I find the study design as well as the processing to be in need of improvement.
As the authors themselves describe in the study limitations, the sample is very small (if gender differences are taken into account, only 5 women were included in the study). Also, the study would have benefited from including the use or fitting of a prosthesis.
Thanks for your suggestions: as we explained in the discussion, both (small sample and prosthesis) are limits of the present study. Recruitment was difficult because many subjects refused to participate in the study.
Also, prosthesis reconstruction should have probably been evaluated at an additional follow-up because many subjects refused prosthesis (mainly due to still present phantom limb sensations or stump pain), and some were not eligible.
The QoL mentioned in the abstract could have been collected more extensively to draw better conclusions and also the statement that the dream content analysis can serve to improve the quality of treatment, I find no further mention in the current text.
Thank you for your suggestion. We substituted “quality of life” with “everyday life” in the Summary and with “pain coping strategies” in the Abstract and in the introduction as we interpreted the quality of life as the ability to deal with pain management and its impact on everyday life.
In the discussion there are wide passages, which should be part of the Introduction.
Thanks for the suggestion: We anticipated the following in the introduction (see also reviewer 1):
“Indeed, in this performance the body acquires a certain organization or style in its relation with its environment. For example, it appropriates certain habitual postures and movements. The body schema is an active, operative performance of the body rather than an image of the existing parts of the body. “It is the body as it actively integrates its position and responses in the environment”
“The self-concept schema depends on “cognitive generalizations about oneself derived from past experiences. Such experiences lead and guide the processing of self-related information acquired from social knowledge. Self-schemas ‘‘reflect one’s core, affect laden assumptions or beliefs about the importance and influence of one’s appearance in life, including the centrality of appearance to one’s sense of self’’
And: Appearance self-schemas are actively triggered by certain internal or environmental events and cues and “Differences have been found among categories in determining self-schema. Men’s and women’s awareness levels of body parts are diverse. For men, body awareness has more permanence compared to women’s, and, in addition, left-handed people seem to be also more certain in perceiving their body senses, compared to right-handed, while the latter showed less body divergence [7] and body awareness may be considered as the conceptualization of body image or the complexity of the principles of body schema, body posture, and body position concerning spatial motions; it’s the basis of personality”.
And: “Dream should not be exclusively defined as a non-conscious electrophysiological state. Neurobiological theories of dreams stated that the meaning of dreams should be considered clearly "transparent" and not elaborated. Hobson affirmed that there are common formal aspects concerning everybody's dreams, and there are some content differences related to age, gender, personality characteristics, and cross-cultural differences.”
We also deleted the following from the discussion: “Consistent with Hobson's view, Calvin S. Hall and Robert L. Van de Castle developed a specific coding system to study dream content. Considering that both innate factors and life experiences may affect the body representation in dreams, they assumed that "the frequency with which a dream element appears reveals the concerns and interests of the dreamer," providing the opportunity "to link dream content with the waking thoughts and behavior of the dreamer"
Also methodologically it is not clear to me how the dream content and the phenomena phantom limb and telescoping are mixed up in this article.
In sleep medicine, the dream analysis and the dreams’ content analysis is still an open chapter. As sleep scientists, we are trying to find alternative ways to explore the mentation and cognitive processes during sleep and what happens while dreaming. We decided to use a scientific approach to the dream content analysis to obtain a possible indicator of the internalization of the body shape modifications (following the Domhoff’s continuity hypothesis). Our previous research focused on mastectomized breast cancer patients, on patients who received a cancer diagnosis and the present study is focusing on amputees who often show these particular phenomena (phantom limb and telescoping)
More detailed analyses of the dream content, possibly with examples, as well as the conclusions drawn from them, would be of added scientific value.
Thanks for the suggestion: Data from dreams’ diaries have been elaborated using the DreamSAT spreadsheet that is a specific tool created to allow the comparison between the research data to normative data and between different series of data. This spreadsheet is designed to facilitate the statistical analysis of dreams that have been coded with the Hall/Van de Castle system of quantitative content analysis. It is not possible to compare specific contents (anecdotal contents) but only categories/groups of data. It is strongly recommended to use this tool to elaborate data previously coded with Hall/Van de Castle system.
Round 2
Reviewer 1 Report
Thank you for addressing my points. I believe that the quality of the manuscript has primarily improved. Here are a few more suggestions:
Line 42: kindly correct «the concept of body «image.»
Line 62-64: Please consider briefly specifying the concept of body permanence and body divergence already in the introduction.
Line 67 – the acronymous PLS should be introduced earlier in the manuscript (at line 77).
- Kindly report in the limitations section the reason for the larger SD at T1 as already explained to the reviewer.
Author Response
Review 1
Thank you for your valuable comments.
Additional corrections are underlined.
Thank you for addressing my points. I believe that the quality of the manuscript has primarily improved. Here are a few more suggestions:
Line 42: kindly correct «the concept of body «image.»
Thank you for your suggestion: We modified the sentence.
Line 62-64: Please consider briefly specifying the concept of body permanence and body divergence already in the introduction.
Thank you for your suggestion: We modified the sentence in the introduction as follows: “For men, body awareness has more permanence compared to women’s, meaning that they are more aware of their body parts. Also, and, in addition, left-handed people seem to be also more certain in perceiving their body senses, compared to right-handed, while the latter showed less body divergence, meaning they have fewer clearness levels of images of different body parts”.
Line 67 – the acronymous PLS should be introduced earlier in the manuscript (at line 77).
Thank you for your suggestion. We modified the sentence and introduced the acronymous earlier.
- Kindly report in the limitations section the reason for the larger SD at T1 as already explained to the reviewer.
Thank you for your suggestion. We added the following in the discussion: Also, the wider SD we obtained at T1 was because we tried to reevaluate the patients during a medical follow-up visit that were differently distributed in a large period of time. It was impossible to reach the patients during an additional visit. This could have missed possible additional changes in the body perception at 4 and 6 months after the amputation.

Reviewer 2 Report
accept in present form
Author Response
Dear Reviewer,
thank you for your valuable comments that improved our paper.